# Content of Pb and Zn in Sediments and Hydrobionts as Ecological Markers for Pollution Assessment of Freshwater Objects in Bulgaria—A Review

**DOI:** 10.3390/ijerph19159600

**Published:** 2022-08-04

**Authors:** Elica Valkova, Vasil Atanasov, Milena Tzanova, Stefka Atanassova, Ivaylo Sirakov, Katya Velichkova, Margarita H. Marinova, Kristian Yakimov

**Affiliations:** 1Department of Biochemistry, Microbiology and Physics, Agriculture Faculty, Trakia University, 6000 Stara Zagora, Bulgaria; 2Department of Biology and Aquaculture, Agriculture Faculty, Trakia University, 6000 Stara Zagora, Bulgaria

**Keywords:** pollution, lead, zinc, accumulation, fish, freshwater sites

## Abstract

The purpose of this review is to describe the contents of Pb and Zn in sediments and hydrobionts as ecological markers for the pollution assessment of freshwater objects in Bulgaria, and the data are compared with other countries and regions. Symmetry was found regarding the levels of Zn in the sediment of the Ovcharitsa and Zhrebchevo dams, which were twice the MAC for arable land (Regulation № 3 of Bulgarian legislation). Symmetry was also observed between the results for Zn and Pb in the studied sediments, and the “favorites” in terms of content were the samples from Zhrebchevo Dam and, especially, from Ovcharitsa Dam. Asymmetry was established in the accumulation of Zn in the livers of carps inhabiting Topolnitsa Dam in comparison with these in Ovcharitsa Dam. A similar asymmetry was observed for lead. The analysis of the muscles and livers of the studied fish showed an asymmetry in the accumulation of zinc, and this process was more intense in the liver. Symmetry was found in the accumulation of Pb in the liver and muscle tissues of the carp from the studied water bodies.

## 1. Introduction

Bulgaria is one of the most dynamically developing countries in eastern Europe and has well-developed industry and agricultural. For this reason, anthropogenic impacts on the hydroecosystems are growing, especially in some industrial regions, such as the Stara Zagora region.

The Stara Zagora region (Figure 1) is located in the central part of the Republic of Bulgaria and is one of the largest territories in the country. It occupies an area of 5151 square kilometers, which is 4.6% of the country’s area, with the largest share of the territory falling in the municipality of Stara Zagora—10,193.62 ha. The smallest is the municipality of Nikolaevo—965.24 ha.

The section covers the northern part of the Thracian lowland, the sub-Balkan field, part of Sredna Gora, and part of the central Stara Planina. The natural complex of temperate climate, diverse and fertile soils, relatively good water potential, and nonobstructive relief has been, is, and will be a favorable factor for the overall development of the area.

The main factors for the development of the agricultural sector in the region are the land, soil wealth, hydromelioration, and water. Many of the water bodies in this area, such as rivers and dams, are used in order to implement industrial activity. The compounds contained in wastewater fall into the organisms of hydrobionts that inhabit these water bodies. The region of Stara Zagora, in which the consortium “Mini Maritsa Iztok” SC (Stock Company) is located, is considered a risk area at all levels for entering the natural food chain—soils, waters, and sediments, as well as plant and animal representatives of the living world [1]. This is the reason for conducting research on the levels of heavy metals in the hydrobionts inhabiting key water bodies in the Stara Zagora region, as well as comparing the data with other countries and regions. Under the action of anthropogenic pressure, significant amounts of harmful emissions are released into the atmosphere, which, falling into water bodies through precipitation, disturb the ecological balance of hydroecosystems. The presence of chemically different toxicants in water bodies is often determined by the inflow of wastewater from the activities of enterprises and agricultural activities carried out in the vicinity of these sites. Contaminants such as heavy metals, paints, organic solvents, pesticides, antibiotics, etc., are found in the incoming wastewater.

Hydroecosystem pollution with Pb and Zn continues to expand and deepen, which seriously damages the safety of hydrobionts and human health. Organisms inhabiting aquatic environments can serve as biomarkers for the presence of various toxic compounds posing not only a serious risk to the consumer, but also leading to large economic losses.

### 1.1. Pollution of Hydroecosystems Depending on Nature and the Impact of Pollutants

Of particular importance for the normal course of all biochemical processes in aquabionts is the preservation of normal indicators of the waters they inhabit.

The pollution of water bodies worldwide is a major problem of our time. The global state of water bodies is an indicator of deteriorating water quality, combined with increased consumption. The main reason for their atypical condition is the intensive development of industry, agriculture, and energy, which leads to water pollution with toxicants, most of which have pronounced carcinogenic and mutagenic effects. Water pollution is carried out by accumulating uncharacteristic substances above a certain concentration, which makes it impossible to use it for raw materials, livelihoods, and habitats [2,3].

An essential factor that has a powerful effect on the pollution of hydroecosystems is anthropogenic activity. As a result of human impact, biological regression is often reached, accompanied by a reduction in species diversity, disruption of the structure and function of ecosystems, and their transition to passive metabolism. The effects of pollution of water bodies with pollutants such as heavy metals depend not only on the amount and duration of exposure, but also on the type of specific water body, the amount of water entering it, the flow rate, the temperature, etc. [4].

Each component of the aquatic environment can be considered a pollutant if levels above certain concentrations lead to negative consequences for the whole system or its individual parts. This concentration value is referred to as the “threshold” or “threshold level”. Interactions between toxicants and the effects of environmental factors on them must be taken into account when setting the content limit.

Sources of water pollution can be natural (lakes, rivers, seas, and oceans), wastewater (industrial and domestic), rainwater, and groundwater.

Depending on their nature and their impact on aquatic ecosystems, pollutants can be:Suspended solids

The presence of such components leads to the deposition of sediments in water bodies and the occurrence of anaerobic conditions. As a result of these processes, the normal life processes of aerobic organisms are disrupted [5,6,7].

Biodegradable organic pollutants

As a result of the activities of the food and biotechnological industries, wastewater is released that, falling into water bodies, causes pollution with carbohydrates, fats, and proteins [8].

Pathogenic microorganisms

The monitoring of water bodies shows that, as a result of people’s household activities, sewage water is infused, which is the main source of pathogens (viruses, bacteria, and helminths), causing a large number of infectious diseases [9,10].

Biogenic elements—N, P, and C [11,12]Through wastewater, compounds containing N, P, and C enter water basins, which causes the eutrophication of water bodies and worsens the taste, smell, and oxygen content of water.Priority pollutants (including heavy metals) [13,14]The summarization of scientific information on the lead and zinc contamination of hydroecosystems in industrial areas on both global and national scales would be useful for ecologists and biochemists. For this reason, the purpose of this review is to describe the contents of Pb and Zn in sediments and hydrobionts as ecological markers for the pollution assessment of freshwater objects in Bulgaria, and the data are compared with other countries and regions.

### 1.2. Standards in Bulgaria and Europe Regulating the Permissible Limits for Heavy Metals in Hydroecosystems

One of the first sectors covered by European environmental policy is water legislation, which includes 25 directives and decisions. A significant part of the provisions of Directive 2000/60/EU (European Union) [15] has been applied to the conditions typical for Bulgaria in the current WA (Water Act) (Promulgated SG No. 67 of 27 July 1999), which was adopted on 28 January 2000. The implementation of the IA ensured the management and protection of water and the health of the population.

In connection with the implementation and supplementation of Directive 2008/105/EC on the setting of environmental quality standards in the field of water policy, on 14 September 2012 Regulation № H-4 for the characterization of surface waters [16] was been adopted. Regulation 4 also regulates standards for the qualities of chemical elements and other substances, including the contents of the heavy metals Zn and Pb.

A no less important component of water bodies is sediment, which performs the role of soil, serving for the development of many plant and animal species. Sediment is a reflection of the history of a body of water and contains information about temporary contamination with various xenobiotics and heavy metals in certain periods of time. This requires sediment samples to be obtained and analyzed during the study of water basins. Of course, in the European and Bulgarian legislation, there are normative documents limiting the quantities of heavy metals, but they are for soil. There are no specific regulatory requirements for sediment, so when interpreting the results of this type of research, the norms for arable soils are used, such as Regulation № 3 (Table 1) of 1 August 2008 in Bulgarian legislation on the norms for the permissible contents of harmful substances in soils [17].

As a result of numerous experimental studies, it has been found that the values of heavy metals in the waters of a given body of water show the current state of the hydroecosystem, which is why a more reliable biological indicator is the body of aquatic organisms. Over time, the tissues and organs of these organisms accumulate heavy elements that, when accumulated above certain levels, disrupt the normal life cycles of aquabionts [18,19,20,21,22].

The lack of a comprehensive, normative document in the European and Bulgarian legislations to regulate the MAC (maximum allowable concentration) and AAV (annual average values) for the contents of heavy metals in all types of organisms inhabiting water bodies makes it difficult to interpret the results of research, including the present. Approved values of the MAC for the contents of most highly toxic heavy elements were determined in the currently inactive Regulation 31 of 29 July 2004 (Table 2) on the maximum permissible amounts of contaminants in food from the Bulgarian legislation [23].

Objects of research in considering the current problem are freshwater fish (*Cyprinus carpio* L.), as representatives of the hydrobionts inhabiting the water bodies in the Stara Zagora region.

Bulgaria’s ecopolicy is aimed at fulfilling the requirements of the European Parliament and the Council contained in EU Regulation № 1881 of 2006, as amended in 2010 for determining the maximum permissible concentrations of certain contaminants in food [24], which corresponds to the currently active Regulation № 5 of Bulgarian legislation (Table 3) that entered into force on 9 February 2015 [25]. These documents regulate quantities of pollutants in order to protect public health by containing them in quantities acceptable from a toxicological point of view. They state that the MRLs apply only to the edible part of the food concerned unless the Annex explicitly states otherwise.

In order to comply with national and European requirements for the contents of heavy metals in the studied aquatic organisms, it is necessary to use accurate methods of analysis (for example atomic adsorption) in determining their quantities in samples of different natures.

Hydrobionts, including all plant and animal organisms inhabiting water bodies, are extremely sensitive to changes in the composition and parameters of aquatic environments, which makes them excellent bioindicators of pollution of all kinds (including heavy metals). These organisms are able to detect the presence of instantaneous peak environmental pollution.

### 1.3. Hydrobionts—Biological Indicators for the Condition of the Aquatic Environment

The interactions between environmental factors, which include individual pollutants (including and heavy metals), and changes in biochemical parameters at the organism or cellular levels lead to a violation of the physiological–biochemical status of organisms, including hydrobionts [4,19,26,27].

The concentrations of heavy metals themselves, which are essential for the state of hydroecosystems, also belong to the hydrochemical indicators of water [28]. Above certain values, these elements have a highly toxic effect on aquatic flora and fauna. Therefore, monitoring and controlling their levels are essential for the environment and human health [19,27]. Heavy metals have the ability to inhibit enzymatic catalytic activity or to promote the production of free radicals and ROS (reactive oxygen species), which oxidize biological macromolecules [29].

Trophic chains are built on the basis of photosynthetic organisms—phytoplankton and higher aquatic plants, followed by zooplankton and higher animal organisms. The composition of water shows only the current state of the water bodies, while the living objects in these systems give a real picture of the ecological and biochemical statuses of these water bodies.

Algae and, in rare cases, macrophytes (higher aquatic plants) are the primary basis of nutrition for the aquatic organisms in marine and freshwater ecosystems. In addition to being primary producers, algae enrich water with oxygen and help precipitate small, organic particles, thus facilitating the process of the self-purification of water [30]. Higher aquatic plants (Water Pepper—*Polygonum hydropiper* L.), as organisms that also start chains in hydroecosystems, accumulate heavy metals contained in water and sediment, so they can be used as reliable bioindicators for this pollution type.

Autotrophs serve as food for many species of freshwater and marine fish. The importance of these aquabionts is not limited to their role as a food source for animals and humans. Being at the top of the food chain, man consumes primarily muscle tissue from hydrobionts and far less caviar and liver (and their products, such as fish oil) of some specific fish species. Fish are distinguished by their ability to accumulate toxicants and serve to assess the ecological statuses of the hydroecosystems they inhabit [31,32]. They serve as good bioindicators for assessing the conditions of water bodies, as they also respond to temporary deviations of the reference values of monitored indicators and, especially, the concentrations of heavy elements. [19,27,33,34].

### 1.4. Biological Species Suitable as Bioindicators for Both Passive and Active Biomonitoring of Water Bodies

Some hydrobionts (for example, the mussel *Dreissena polimorpha (Pallas, 1771)*) are powerful biological filters of marine and freshwater basins, which purify the water and provide a better environment for the development of other aquatic hydrobionts [35,36]. The abilities of mussels make them reliable bioindicators for both the passive and active biomonitoring of water bodies [37,38].

Common carp (*Cyprinus carpio* L.) and crucian carp (*Carassius* spp.) are very convenient as fish species for the indication of heavy elements. Their ubiquitous distribution, their resistance to low temperatures, the heterogeneous composition of their environments, and their occupation of accessible habitats are the reasons why representatives of this genus are often used as objects of research related to determining the contents of pollutants in aquatic ecosystems [27,39].

### 1.5. Accumulation of Pb and Zn and Their Effect on the Organisms of Hydrobionts and Humans as Consumers

One of the main causes of degenerative processes in the environment is heavy metal pollution [40,41]. The microelements (such as Fe, Cu, Zn, etc.) present in natural waters are in the form of organic and inorganic salts. There are three states in which heavy metals can be in an aqueous medium: dissolved, colloidally dispersed, and suspended. In small quantities, zinc is absolutely necessary for the vital activity and reproduction of all living organisms. Above certain values, however, lead and zinc have detrimental effects on the ecological balance and diversity of aquatic organisms [4,42,43,44]. Due to their toxic effects, these metals disrupt the normal course of biochemical processes in aquatic inhabitants of natural and cultural ecosystems [19,45]. By affecting aquatic ecosystems, they also affect terrestrial ones, as they spread to all trophic levels.

The ionic forms of these heavy metals enter the bodies of hydrobionts, and concentrated solutions of their salts have a burning effect and damage the respiratory organs. Salts are contained in sediment, silt, and waste masses. It has been found that a large part of the inorganic compounds of metals enters the bodies of fish and aquatic invertebrates through food. Soluble dissociating salts and organometallic compounds penetrate their bodies through the gills and skin [46].

The main depots of accumulation of heavy elements are the liver, muscle tissue, and eggs of fish, mussels, and sea snails. A large number of heavy metals (including Pb and Zn) easily enter into the larvae and eggs [20,47,48,49,50]. Gametes, juvenile larvae, and small fry fish are the most sensitive to changes in the ecological and biochemical statuses of hydroecosystems [20,51].

Heavy elements (Pb and Zn) have been found to accumulate mainly in the liver, gills, and gonads (ovaries and testes) and, to a lesser extent, in the muscle tissue of fish and mussels [28,31,32,52,53]. The liver, as the organ with the highest degree of metabolic load, has the ability to accumulate high levels of heavy metals over time and to serve as a reliable biological marker for the pollution of aquatic systems [4,54,55]. The accumulation of these elements is also observed in the genitals of aquatic organisms, but to a lesser extent. In most cases, small amounts of these metals accumulate in the muscle tissues that are most often used for sustenance.

In dilute solutions, heavy metals penetrate the body and disrupt the permeability of the plasma and intracellular membranes, oxidizing their major components and inhibiting their most important property—selective permeability [56,57,58].

Numerous studies have shown that heavy metals bind to thiols (sulfhydryl) [59,60] and amino groups of proteins [46]. At high doses, these elements induce lipid peroxidation [55,58] and lead to the disruption of ion transport across cell membranes [61]. In fact, lipid oxidation is a consequence of oxidative stress caused by high levels of heavy metals and the formation of ROS. Obtaining ROS can cause not only oxidation of proteins and lipids, but also activation of the expression of genes encoding antioxidant enzymes, increased concentrations of agents that eliminate reactive oxygen species, and changes in cellular redox status [62].

Of particular importance among the trace elements is zinc, which is present in the composition of every living cell. It is a normal component of important enzymes, such as carbonic anhydrase and alcohol dehydrogenase. About 90% of the total zinc in the body is concentrated in muscle tissue and bones. Zn levels in the liver, gastrointestinal tract, skin, kidneys, brain, lungs, prostate, and other organs are significantly lower [63].

This metal plays an essential role in maintaining the integrity of the cell membrane, regulates the action of insulin and blood sugar concentrations, and has a direct effect on the development and maintenance of the body’s immune system. Zinc is also needed to maintain bone and tooth mineralization and normal wound healing. As a component of more than 200 metalloenzymes and other metabolic compounds, zinc provides stability to biological molecules such as DNA (deoxyribonucleic acid) and biological structures such as membranes and ribosomes [64,65,66,67].

High concentrations of zinc have a highly toxic effect on living organisms, including aquatic organisms. Zinc ions (Zn^2+)^ cause structural changes in the major classes of lipids and significant decreases in the levels of malonic dialdehyde and diene conjugates [68].

Zinc is a metabolic antagonist of cadmium, thanks to which it manages to reduce to some extent the toxic effect of Cd. Anemia is usually seen in mammals as a result of sublethal zinc intoxication. In fish (trout), the effect is necrosis of the liver tissue [18].

Zinc, like other heavy metals, accumulates in organs such as the liver, spleen, kidneys, and gonads [69,70,71]. In humans, zinc intoxication can lead to various types of anemia, as well as damage to the lungs, skin, reproductive system, and nervous system [72].

Lead is characterized by multifaceted toxicity and widespread distribution. This metal has extremely low chemical mobility. It exists in the form of metallic lead, inorganic ions, and salts [73]. The main sources of Pb in the aquatic environment are ore mining and the production of batteries, chemicals, and dyes.

Lead is distinguished by the fact that, not only does it play a significant role in biochemical aspects, but it has a powerful toxic effect on organisms. Over 90% of this toxicant is transported by erythrocytes. In high doses, this metal exhibits embryotoxic effects [74] and causes the disorders of hematopoiesis and erythropoiesis [75], neurotoxic effects, and binding to thiol groups [59]. The effect of Pb is also expressed in lesions of the digestive, reproductive, and nervous systems.

High bioaccumulations of Pb have been found mainly in mollusks and other invertebrates [76]. The degree of absorption in fish increases with increasing pH and decreases with increasing hardness. The main depots of lead deposition are mainly parenchymal tissues, such as the liver and kidneys, and to a lesser extent the bones of aquatic organisms. High levels of this metal have also been reported in fish gills, which is evidence that this element is absorbed by the respiratory system. Increasing its concentration at the ecosystem level leads to a reduction in biodiversity and disruption of the development of ichthyofauna in water bodies [18].

Fish and mussels, having the ability to accumulate heavy metals, can be a natural source for Pb intoxication in the human diet [77]. Children are particularly sensitive to this metal due to their faster growth rate and metabolism, which can cause critical effects on the developing nervous system [78].

Scientific research on the toxic effects of heavy metals usually studies the effects of metals individually, but hydrobionts are exposed to the complex effects of heavy elements common in nature.

From a scientific point of view, data on the antagonistic relationships between heavy metal ions are of greater interest. In this regard, studies have shown that the intake of larger amounts of Cu, Zn, and Co plays a protective role against the effects of lead [29]. Zinc reduces the actions of Pb and Cu in relation to its action, causing changes in the structure and function of erythrocyte membranes. In the absence of Fe, an increase in Pb resorption was observed [79].

The concentrations of heavy metals found in the waters of different types of hydro-ecosystems show the current states of hydroecosystems, which is why aquatic organisms are a more reliable indicator of their statuses. The conditions of water bodies can be realistically assessed only by monitoring the studied indicators in different organs of hydrobionts, especially since there are specific biochemical mechanisms for the accumulation of certain heavy metals and xenobiotics in tissues and gametes [4,18,20,27].

Hydrobionts are organisms that are essential for the survival of mankind due to the fact that they are a natural link in the existing food chain with humans as the end link. Used as food, they are a source of extremely important ingredients, such as essential amino acids, essential fatty acids, and vitamins that are important for metabolism [80]. However, at the same time, they can cause toxic levels of xenobiotics and heavy metals.

The path of transport of heavy metals in hydroecosystems starts with the water, followed by the sediment of water bodies, and subsequently, they enter plants and other aquatic organisms. These effects are accompanied by symmetrical–asymmetrical interactions between these elements.

## 2. Contents of Pb and Zn in the Sediment of Freshwater Water Bodies

Heavy metals contained in the form of organic and inorganic structures do not have the ability to be eliminated, which is why in the presence of high concentrations they accumulate in the sediments of water bodies. Precipitation depends on the water flow rate and the particle size. This process leads to decreases in the concentrations of various pollutants (including heavy metals) in aquatic environments. The accumulation of metals in sediments is controlled by sorption and ion exchange, which are also processes occurring between metals in the solution and the sediment.

According to a study related to the content of heavy metals in river sediment, Facetti et al. [81] reported high levels of metals after the direct disposal of waste from the fertilizer and textile industries. The reported symmetry regarding the larger amounts of metals registered during the summer and winter seasons was most likely due to the lower flow rate and easier accumulation of these elements in the mud. 

When studying the content of heavy elements in the period of May–September 2007 in sediments along the Chiprovska Ogosta River, it was found that, mainly, amounts of Zn, Ni, and As were accumulated, as well as larger amounts of Mn, Fe, Co, and others [82]. Pb and Cu were found in higher percentages in the residual phases of the sediment. The main mechanisms through which the sedimentation of metals in these water bodies took place were adsorption, cation exchange, and coprecipitation. This study was in accordance with the generally accepted thesis that, in the sediment of water bodies, there is concentrated a large part of metals, the concentrations of which increase over time due to the process of accumulation.

In the period of 2009–2013, a study of the amounts of heavy metals in the Middle Iskar Cascade, which covers 33 km of the river sector in the middle part of the Iskar River, northwestern Bulgaria, which is part of the Danube basin, was conducted [83]. In the analysis of the obtained data, it was found that, in 2009, the sediments at the beginning of the cascade were characterized by higher concentrations of As, Cu, Pb, and Zn. The registered levels exceeded the MAC for soils by 0.8 to 3 times. In 2010, a slight increase in the content of these metals was found, but at the end of the cascade. As a result of this study, it could be concluded that the combination of high organic content, low grain fraction, increased sedimentation and water retention, and compounds in the dams of this cascade had a large effect on heavy metal pollution and increased environmental risk.

Alia et al. [84] reported high levels of Cr in the sediment of the Karnaphuli River in Bangladesh due to the direct disposal of untreated waste from the oil, fertilizer, and textile industries [82,83,84]. Higher concentrations were also found in the lead levels in this study, with the amounts of Pb and Cd metals testing higher in summer and winter due to the lower water flow, which could help with the accumulation of heavy metals in sediment [85,86]. There is a constant movement of water in the rivers, which complicates the sedimentation process. The mean Cd concentration was 1.51 mg/kg in the summer and 2.50 mg/kg in the winter. The average Pb values measured in the river sediment were 38.33 mg/kg and 49.04 mg/kg, respectively, during the summer and winter seasons.

Global studies and high levels of environmental pollution with various types of xenobiotics are the reason why many studies have been conducted in Bulgaria in connection with establishing the contents of heavy metals in the sediments of freshwater water bodies and the degrees of accumulation of these elements, as well as the factors on which accumulation depends.

The results of the global studies cited above and on the territory of Bulgaria have determined the need for monitoring the quantities of heavy metals in the sediments of the freshwater water bodies in the Stara Zagora region. A large part of the research interpreted in this review is related to the implementation of an international project titled as the “Assessment, reduction and prevention of air, water and soil pollution in the Stara Zagora region” No. 2008/115236. We conducted our own studies, which traced the concentrations of some heavy metals (including zinc and lead) in the sediments of Zagorka Lake, Ovcharitsa Dam, Opan Dam, Pastren Dam, Gita Dam, Zmeevo Microdam, Sazliyka River, Tundzha River, Yagoda Village, Zhrebchevo Dam, the Nikolaevo checkpoint, the Zhrebchevo Dam wall, and Tundzha river after the Zhrebchevo Dam, all located in the risk region of Stara Zagora.

### 2.1. Content of Zn in the Sediments of Freshwater Water Bodies in the Territory of the Stara Zagora Region, Bulgaria

As a result of the conducted biochemical analysis in the course of our studies, clear differences were found in the determination of the heavy metal content of Zn in the sediments of the studied water bodies. The highest levels were registered in the queue of Zhrebchevo Dam and Ovcharitsa Dam, twice exceeding even the MAC in arable land [17]. It was obvious that Ovcharitsa Dam, due to its location (near the TPP (thermal power plant) of Maritsa East-2), absorbed a number of heavy metals disposed of through harmful emissions from production. Large amounts of heavy metals are also observed in the soil of Opan Dam, probably for similar reasons. However, this sediment deposition could be due to both rare peak loads and weaker but permanent essential contaminants [4].

The four points along the Tundzha River turned out to be especially useful for the overall interpretation of the results. The low levels of heavy metals near the village of Yagoda and the high ones near Nikolaevo, which were again very low both at and after the wall of the Zhrebchevo dam, suggested at least two facts:Heavy metal pollution was in the section between the village of Yagoda and the town of Nikolaevo, probably from the recent mining activity in the region of Tvarditsa;Zhrebchevo dam served as a precipitator of heavy metals (and probably other pollutants), thus purifying the Tundzha River.

### 2.2. Content of Pb in the Sediments of Freshwater Water Bodies in the Territory of the Stara Zagora Region, Bulgaria

The already established trends were also reported for the heavy metal lead (Pb). Symmetry was observed between the results for Zn and Pb in the studied sediments, as again the “favorites” in terms of content were the samples delivered from Zhrebchevo Dam and, especially, from Ovcharitsa Dam. The concentrations of lead were two times lower, but relatively high compared to the relatively purest samples (wall of Zhrebchevo Dam), in the Sazliyka River, Zagorka Lake, and Gita and Opan Dams.

All the sediment samples tested did not exceed the limit provided for in Regulation № 3 of 1 August 2008 on the standards for the permissible content of harmful substances in lead (Pb) for soil in arable land at pH from 6 to 7.4, which was valued at 100 mg/kg. However, large amounts of Pb were observed in the analyzed sediment from the Tundzha River at the Nikolaevo point (42.96 mg/kg). The sampling point was the mouth of the Tundzha River, through which it flows into the Zhrebchevo Dam. In addition, this area of the river is close to the city of Nikolaevo, which is why it is subject to strong anthropogenic influence. High concentrations were also measured in the samples from the Sazliyka River (25.38 mg/kg). The sediment taken from the Tundzha River in the village of Yagoda was characterized by low concentrations of lead (10.3 mg/kg) due to the high speed of the water currents. There was a higher value in the samples from Zhrebchevo Dam—Nikolaevo point and a gradual decrease in the levels of lead at the points of the Zhrebchevo Dam—wall and Tundzha River—after Zhrebchevo Dam [4,19].

## 3. Content of Pb and Zn in Algae and Higher Aquatic Plants in Freshwater Bodies and Their Role as Ecological Indicators

Microalgae, multicellular algae, and higher aquatic plants are autotrophs that are located at the base of each food chain in hydroecosystems of different types. The importance of these organisms is not limited to the nutrition of many species of invertebrates and vertebrates, but it is also related to their ability to extract organic and inorganic substances from the water and sediment of the water body they inhabit. The present qualities of these organisms have led to numerous studies on the susceptibility of aquatic organisms to toxicants (including heavy metals) and their use as markers for the purity of hydroecosystems.

Studies by Abo-Rady [87] and Mortimer [88] have confirmed that aquatic macrophytes and some algae can be used for bioindication to assess the presence of selected heavy metals in aquatic ecosystems. They are used as bioindicators for environmental assessment due to their distribution, size, longevity, presence in places of pollution, ability to accumulate metals to a satisfactory degree, and easy identification [89,90,91,92].

Studies on the ability of algae and higher aquatic plants to accumulate heavy metals and their role as suitable bioindicators for pollution with this type of pollutant have given grounds for studies in the Stara Zagora region, Bulgaria, due to the high level of anthropogenic pressure to which it is subjected.

### 3.1. Pb Content in Algae and Aquatic Plants Inhabiting Freshwater Water Bodies in the Stara Zagora Region, Bulgaria

Lead does not have a positive effect and does not participate in the biochemical reactions that are characteristic of plant cells. Even in small doses, its effects are very negative for organisms. Determining its levels in these types of samples is essential for characterizing the water bodies we studied.

The analysis of the results obtained in determining the amounts of lead in the samples of algae and water pepper in the course of our studies showed that this element was actively absorbed from the soil through the roots of water pepper, and samples of this aquatic plant had the highest values of lead (10.57 mg/kg). It could be concluded that water pepper acted as an “accumulator” of some heavy metals, and this had an important practical application for cleaning the soil in some polluted shallow water bodies.

Relatively high values of Pb were also found in the samples of algae from the Sazliyka River (9.26 mg/kg) and the Tundzha River after the wall of the Zhrebchevo Dam (8.02 mg/kg). It was assumed that the reason for the observed symmetry in the obtained results was that, in the studied period, this element was in larger quantities in the specific hydroecosystems. Algae from the Pastren Dam (0.41 mg/kg) were characterized by the lowest values [4,19].

### 3.2. Content of Zn in Algae and Aquatic Plants Inhabiting Freshwater Water Bodies in the Stara Zagora Region, Bulgaria

The microelement zinc has a strong influence on the activity of many enzymes. This metal is also involved in the regulation of sugar metabolism and protein synthesis. It occurs in cell membranes and ribosomal structures. Together with copper, it forms superoxide dismutase (SOD) [93]. It has been found that the use of brown macroalgae as a natural ion exchanger allows the sequestration of positively charged metal ions, such as copper, zinc, cobalt, cadmium, lead, and others [94].

During research under a Norwegian program [4], the highest levels of Zn were found in the samples delivered from three water bodies—the Ovcharitsa, Opan, and Zhrebchevo Dams (Nikolaevo checkpoint). The observed symmetry was clearly expressed. It is interesting that the algae obtained for the study from the wall of Zhrebchevo Dam were characterized by 53 times lower levels of this heavy metal compared to those from the tail of the dam, as well as 57 times lower than those of Ovcharitsa Dam. Apparently, the asymmetry “sedimentary effect” was observed for Zn at Zhrebchevo Dam, which is generally valid for sediment, but in this case, it was also observed in algae [4].

The conducted research on the contents of some heavy metals in plant samples further clarified the ecological situation in the studied water bodies in the Stara Zagora region during this period, as important relationships were found between the levels of heavy metals contained in aquatic plants and algae.

In aquatic ecosystems, there is a complex network of relationships that are not inferior to those in terrestrial ones. There is a huge variety of living forms in the fresh and salty waters of the Earth, some of which are still unknown. The presence of many plant and animal species determines the complex network of food relationships there.

An essential link in the food chains in water bodies is all the known and unknown species of fish. These aquabionts are extremely sensitive to the contamination of water bodies with potential xenobiotics and heavy metals [4,95,96,97].

An extremely suitable object for research that is characterized by widespread distribution is the common carp (*Cyprinus carpio* L.), a freshwater fish. This species has proven bioindicators abilities, is an omnivorous fish, and inhabits even swampy ponds. In general, fish, as aquatic organisms, tend to accumulate heavy elements [98].

According to data published in the scientific literature worldwide, the accumulation of heavy metals in fish is concentrated in organs with active metabolisms, such as the liver, pancreas, gonads, and gills, which are respiratory organs [54,99]. To a lesser extent, heavy elements accumulate in the muscle tissue of these aquatic organisms, which are used as a source of nutrition by humans as well.

During our study, the contents of heavy metals in the livers of common carp from water bodies located in the Stara Zagora region were initially studied.

## 4. Contents of Pb and Zn in the Livers of Common Carp (*Cyprinus carpio* L.) from Freshwater Bodies

The liver is undoubtedly the first in terms of metabolic load in the bodies of vertebrates (especially fish). It is considered a central organ of metabolism, and as such, is characterized by the fact that significant amounts of blood and lymph pass through it, supplying components of exogenous and endogenous origin. In the literature, the liver is often compared to a “universal biochemical laboratory”, and hepatocytes are defined as “accumulators” of substances and energy needed for later needs of the body. As an organ with a proven capacity for the bioaccumulation of various toxicants (including heavy metals), fish liver can be and often is used as a reliable biological marker for the presence or absence of high-toxicity pollutants in different types of hydroecosystems.

It is well-known worldwide that the liver is a central storage organ in the body, in connection with which many studies have been conducted over time related to the accumulation of heavy metals in it.

In a study on the accumulation of some heavy elements in common carp (*Cyprinus carpio* L.), Vinodhini and Narayanan [100] found that significant amounts of Cr, Ni, Cd, and Pb accumulated in this organ. Asymmetry was observed between the liver and other organs and tissues, with the liver of carp having the ability to accumulate these elements over time and, thus, prevent large amounts of them from entering other vital organs and muscle tissue that could be consumed by animals and man.

In a study conducted in 2012 by Yancheva et al. [101], they also found high levels of some heavy metals (including Pb and Zn) in samples from the internal organs of common carp and, in particular, in the livers of these fish caught from Topolnitsa Dam through research period. The degree of bioaccumulation in this organ was determined to be high. The concentrations recorded in the liver of this species of fish during the active season in the spring were as follows: Pb—2.4 mg/kg; and Zn—109 mg/kg. The observed data showed high levels of lead and zinc, which most likely accumulated in the fish bodies due to high metabolism during this part of the year. More interestingly, the amounts of zinc were very high, although this metal is an active participant in metabolism and is widely used in cellular metabolism. The metals of copper and nickel, which also belong to the essential elements, were also registered at high levels, but lower than those of lead and zinc, which was completely understandable given their participation in the structure of enzyme systems important for metabolism. The established symmetry in high levels of heavy metals clearly showed the accumulative capacity of the liver, as well as the importance of its use as a biological marker of pollution.

Georgieva et al. [102], in a study on liver samples of common carp inhabiting the Topolnitsa Dam, in 2013 certified the presence of large quantities of the already mentioned heavy metals of copper, nickel, lead, and zinc. During the spring season, the following concentrations of these elements were measured: Cu—15.7 mg/kg; Ni—1.3 mg/kg; Pb—2.39 mg/kg; and Zn—108.7 mg/kg. The liver is a vulnerable organ for prolonged exposure to heavy metals from water sources or various foods and is, therefore, one of the main target organs in toxicological studies [103]. The results obtained in this study proved once again that, in fish, the liver is the main depot for the bioaccumulation of heavy metals, and this statement is consistent with the research of Shinn et al. [104] and Poleksić et al. [105]. The observed symmetry in the high concentrations of these elements in the livers of fish may be related to the processes of hematopoiesis and detoxification, as well as to the antioxidant defense system and the excretion of metals from the body [106].

The research of Alvarado et al. [107] was also related to the analysis of samples of carp liver and muscle tissue and confirmed the presence of higher levels of the metals studied by them in the livers of the captured specimens. The levels of Cu (mg/kg) recorded in the livers of fish in this case ranged from 6.44–47.81; for Zn (mg/kg), in the range 38.44–265.00; and for Cd (µg/kg) and Pb (µg/kg), 13.25–472.40 and 9.70–76.38, respectively. Obviously, the accumulation in this organ was significant, i.e., its ability to accumulate toxicants such as heavy metals was confirmed, preventing their accumulation in other organs and tissues.

Other evidence for the bioaccumulative and metabolic abilities of the liver was obtained in the course of our study [4]. An analysis of liver samples from carp inhabiting some of the water bodies in the Stara Zagora region in mentioned years was performed. The aim of the study was to trace the dynamics of the studied heavy metals over time and to establish their impact on the presence or absence of persistent pollution in the waters of the studied water bodies.

The large-scale study we conducted covered the water bodies of Zagorka, Ovcharitsa, Opan, Chirpan, Zetyovo, Pastren, Koprinka, and Zhrebchevo. In 2011, the objects of the study were the dams of Ovcharitsa, Opan, and Pastren, which are located near the TPP of Maritsa—East 2, which is why they were considered to be subject to strong anthropogenic impact. The data obtained from the analysis for determining the contents of heavy metals in the samples from Ovcharitsa Dam in 2010 and 2011 gave grounds to conduct an additional study in 2014. As is well-known, the accumulation of heavy metals in fish is concentrated in the liver and, to a much lesser extent, in the muscle tissues of these aquatic organisms. In connection with this fact, the study initially examined the contents of heavy metals in the livers of carp from water bodies located in the Stara Zagora region.

### 4.1. Content of Zn in the Livers of Common Carp from the Study Water Bodies in the Territory of Stara Zagora Region, Bulgaria

Zinc is also included in the group of essential elements. Its presence is absolutely necessary for normal metabolism processes. In small amounts, Zn is needed as an essential component of a number of proteins involved in carrying out the functions of the nervous and endocrine systems [108].

During our study in 2010, it was found that the levels of Zn in the samples of carp liver were not significantly different, and the quantities followed the previously established trend for the levels of the elements we studied (including lead).

The chemical analysis of the liver samples showed that the hydrobionts that inhabited the Zagorka, Ovcharitsa, Zetyovo, and Chirpan water bodies were again distinguished by the highest values of Zn. The quantities registered in the samples from Zagorka Lake, Ovcharitsa Dam, and Zetyovo Dam were significantly higher than the norms defined in Regulation 31 regarding the MAC for Zn (50 mg/kg). In European Regulation 1881, as well as in the current Regulation 5/2015 [25] and the present Bulgarian legislation, the limits for this element are not specified. In the period of 2009–2013, a study on the amounts of heavy metals in the Middle Iskar Cascade, which covers 33 km of the river sector in the middle part of the Iskar River, northwestern Bulgaria, which is part of the Danube basin, was conducted

In 2010, our study found values of Zn that were clear evidence of elevated levels of this metal in most of the water bodies during the above study period. The established symmetry in the results was ubiquitous.

In a study in 2011, the highest value was measured in the livers of common carp from the Pastren Dam. All the registered concentrations in this period did not exceed or approach the requirements of Regulation 31 of the Bulgarian legislation [23] (24.4% below the regulated MAC). The current Bulgarian and European normative documents do not set limits on the content of zinc in meat of, even more so, in the livers of the studied aquatic organisms.

Zinc levels recorded in 2011 in the samples from Ovcharitsa Dam were much lower than the ones measured in 2010, as the difference was over 30 units (53.3%). The observed asymmetry in the results of the two studied years showed that, apparently, in a past period of time there was the presence of instantaneous pollution. Most of the biochemical reactions of the body take place in the liver, which determines its ability to seal that moment by accumulating excess zinc. The data obtained confirmed the fact that this organ acted as an indicator of the purity of the hydroecosystem inhabited by the studied aquatic organisms [4,54].

The concentrations of Zn measured in the livers of hydrobionts in 2010 and 2011 gave grounds for conducting our own study in 2014 to clarify the state of the hydroecosystem of Ovcharitsa Dam (Figure 2). The analysis showed that the relatively low levels of Zn in the waters of Ovcharitsa Dam were confirmed. A slight increase in the arithmetic mean value was registered by about four units compared to 2011. The reported variation in zinc levels did not change the final result, in which the established value was lower than the norms in Regulation 31 by 38%. As mentioned, there are no restrictions on the concentrations of this metal in food in the current Bulgarian and European documents [4,54].

In the same time range (2010–2014) for the studies of Yancheva et al. [101], high levels of Zn (109 mg/kg) were also recorded in carp liver samples taken from Topolnitsa Dam.

The quantities of this heavy metal were compared with the results obtained in our study from the period of 2010–2014, which showed that the accumulation in common carp inhabiting the Topolnitsa Dam in 2012 (Figure 2) was significantly higher and twice-exceeded the requirements of Bulgarian legislation Regulation 31 [23]. This result indicated the presence of large amounts of zinc in the whole bodies of the studied fish, not only in accumulative organs, such as the liver. Most likely, at that time the levels of Zn in this hydroecosystem were significantly high.

The observed symmetry regarding the Zn contents in the studied samples from the Ovcharitsa and Topolnitsa dams was clear evidence of the ability of the liver to be a detoxifying and accumulating organ in fish, as well as its possibility to be used as a biochemical marker for aquatic pollution.

Through their research, Agah et al. [109] confirmed the fact that zinc, like other heavy elements, accumulated mainly in the liver tissue of fish.

### 4.2. Content of Pb in the Liver of Common Carp from the Studied Water Bodies in the Territory of Stara Zagora Region, Bulgaria

Lead also belongs to the group of highly toxic metals, and it not only contributes to the normal course of biochemical reactions in aquatic organisms, but even in small doses it has a powerful toxic effect. The accumulation of Pb in fish tissues leads to oxidative stress due to excessive production of ROS (ROS—reactive oxygen species). Oxidative stress during Pb exposure leads to synaptic damage and the malfunction of the neurotransmitters in fish, i.e., it causes neurotoxicity. In addition, exposure to Pb affects immune responses in fish, disrupting their normal course [110].

The data showed that the situation regarding the content of Pb in the livers of fish inhabiting the water bodies that were the subject of study in 2010 was extremely worrying. All the tested samples were many times higher than the MAC for this dangerous heavy metal, as regulated in Regulation 31 [23]. In the current Regulation № 5 of 9 February 2015 [25] from the Bulgarian legislation and, respectively, European regulation 1881, there are no requirements regarding the MAC for lead in the liver of fish, as this organ does not belong to the edible part of the aquatic organisms.

Laboratory tests performed on the liver in 2010 clearly showed record levels of lead in the samples from the water bodies of Opan, Ovcharitsa, Koprinka, Pastren, and Zagorka. The quantities registered in the samples from the Opan, Koprinka, Pastren, and Ovcharitsa dams were ten times higher than the MAC of 0.2 mg/kg, asregulated in the normative documents for this period. Results similar to those reported for the amounts of lead in the liver using the same study object were also obtained from Yancheva et al. [101]. The observed symmetry in the results of the two studies confirms that the liver of fish, accumulating Pb, reports the probable presence of large amounts of heavy metal in the studied ecosystems during the above period. Due to its ability to be a major depot of heavy metals in the bodies of aquatic organisms, this organ accumulates the available amounts of lead [101,111]. The observed phenomenon was completely understandable, as the nutrients of Na, K, Ca, Mg, Zn, Cu, and others were included in the bodies and, due to their active participation in metabolism, were consumed. The highly toxic metals to which lead belongs cannot be consumed and accumulate in the liver and other active tissues [4,54].

According to conducted studies, the presence of such high concentrations of lead can result in disorders in the size and content of erythrocytes, the stomach, the intestines, and the reproductive and nervous systems of the studied aquatic organisms [82].

The reported levels of lead in the liver samples of carp inhabiting the studied water bodies in 2011 were relatively low and met the requirements of Regulation 31 [23]. The highest levels were measured in the samples from Ovcharitsa Dam (42% below the MAC regulation in this period). This concentration was ten times lower than the one measured in 2010. Most likely, it was a matter of momentary pollution in the above period. The accumulation of toxicants of this kind can be explained by the fact that each organism personally accumulates a given pollutant. The process of accumulation depends on many varied factors of the external and internal environment—ecological, biological, and genetic [112].

The sharp decline in Pb levels in the period of 2010–2011 created the need for further studies on the content of this metal in the livers of the studied fish species, which were conducted in 2014.

The amounts of Pb that were measured in the samples of carp liver from Ovcharitsa Dam in the studied period of 2014 eloquently proved the absence of this extremely toxic metal in that specific hydroecosystem (Figure 3). Symmetry was observed regarding the lead concentrations in the liver samples of these fish in 2011 compared to 2014. The results obtained confirmed the thesis of possible momentary pollution in 2010, which inevitably affected the levels of lead in the livers of the studied aquatic organisms. There was an increase in the levels reported in 2014 by 25.4% compared to 2011, but the results of both years were several times lower than the values measured in 2010. The registered values for this period were 20% below the current norms [4,54].

In a study by Yancheva et al. [101] conducted in 2012 with carp specimens inhabiting Topolnitsa Dam, very high levels of Pb were found, exceeding by 10 times the normative requirements set out in Regulation 31 (0.2 mg/kg), which set standards for the consumption of whole fish. In the current normative document—Regulation № 5 of the Bulgarian legislation [25] and Regulation 1881 of the European Parliament—there are no such restrictions [24].

Compared to our data from the research on liver samples from carp inhabiting Ovcharitsa Dam in the period of 2010–2014 (2010—1.68 mg/kg), these results were several times higher (Figure 3). The presence of such doses of lead in the liver of these carp definitely indicated the presence of such in other metabolically active organs and muscle tissue of the same individuals.

## 5. Content of Pb and Zn in the Muscle Tissue of Common Carp (*Cyprinus carpio* L.) from Freshwater Water Bodies

The muscle tissue is an important part of the bodies of hydrobionts, both from a physiological–biochemical perspective and from a dietary point of view, unlike the liver, which occupies a central place in the metabolism. Humans, as the final link in the food chain, consume primarily muscle tissue from aquatic organisms and far less caviar and liver (and their products, such as fish oil, etc.) of some specific species of fish. The legal restrictions concerning the levels of some of the heavy metals studied in the 2010, 2011, and 2014 surveys were regulated in Regulation № 31 [23], as enforced at that time, which is why the data obtained were interpreted in relation to this normative document. This normative document set permissible limits only for highly toxic heavy metals (lead, copper, cadmium, arsenic, etc.), and for elements such as iron and manganese, no maximum permissible concentrations were provided, probably on the presumption that they are useful elements and that human organisms suffer from deficiency rather than excess. The situation is similar to the currently active Regulation № 5 of 9 February 2015 of the Bulgarian legislation [25], which in practice applies Regulation № 1881 of the Commission of the European Communities of 19 December 2006 with an amendment from 2010 for determining the maximum permissible concentrations of certain contaminants in food [24]. However, the lack of restrictions in the current regulations on the contents of certain heavy metals in food makes it difficult to interpret the data obtained. Since hydrobionts are an extremely suitable and predictive marker showing the anthropogenic load of water bodies for heavy metals, the analysis of all of the studied metals becomes possible and interesting. Although waters in most cases are characterized by relatively low levels of heavy metals, they accumulate in the bodies of the aquatic organisms inhabiting them through specific biochemical mechanisms other than simple diffusion.

The content of heavy elements in the muscle tissue of freshwater fish is of interest to scientists in Bulgaria due to the importance in its consumption by animals and humans.

During her study, Velcheva [103] compared the levels of zinc (Zn) in the organs and muscle tissue of freshwater fish inhabiting the Kardzhali and Studen Kladenets dams for three consecutive years. For the first year, in the muscle tissue of carp inhabiting the Studen Kladenets Dam, the highest values were reported in the autumn (28.48 mg/kg), which compared to the amounts in the livers of the same fish, were extremely low (Zn for liver in the same period was 201.21 mg/kg). The amounts of zinc in the muscle tissue of these aquatic organisms in the second year were highest in summer (28.98 mg/kg). At the same time, carp liver levels were 458.70 mg/kg. In the third year, low levels of zinc were reported, but the accumulating role of the liver and the presence of small amounts of this metal in the muscle tissue of the studied fish were still clearly visible [103].

The study by Zhelyazkov et al. [96] confirmed the thesis that the muscle tissue of fish (including freshwater) accumulates much smaller amounts of heavy metals due to the retaining role of organs, such as the liver, gills, gonads, and others. In this study, conducted in the period of March–October 2016, the amounts of the heavy metals of cadmium (Cd), nickel (Ni), lead (Pb), and zinc (Zn) in skeletal muscle tissue (*Abramis brama*, *Linnaeus*, *1758*), Prussian carp (*Carassius gibelio*, *Bloch*, *1782*), vibra (*Vimba vimba*, *Linnaeus*, *1758*), and maple (*Leuciscus cephalus*, *Linnaeus*, *1758*) inhabiting the Zhrebchevo dam at that time were measured. The highest levels of the highly toxic metal of cadmium in this analysis were determined for bream at 0.045 mg/kg. Other fish species (including carp) had a low Cd content in the muscle tissue (0.01 mg/kg). These results are symmetrical with data from the studies of Peycheva et al. [113], who found cadmium concentrations in the range of 0.020–0.046 mg/kg in representatives of the Carp family from Lake Mandra, as well as those reported by Yancheva et al. [101] for Topolnitsa Dam. In all these studies, the permissible norms for Cd, regulated in Regulation 31 of Bulgarian legislation [23], as well as in European regulation 1881 and the now-active Regulation № 5, were not exceeded. The other metals analyzed were in quantities significantly smaller than the requirements set in the regulations. The research of Yancheva et al. [101] and Peycheva et al. [113] has also confirmed the presence of low levels of heavy metals in the muscle tissue of freshwater fish at the expense of accumulative organs, such as the liver and others. There was an asymmetry between the contents of these elements in the livers and muscle tissues of the studied fish species, which is important from dietary and biochemical points of view, as the meat of these aquatic organisms is a source of protein and useful fats for animals and humans.

The ability to accumulate large amounts of toxicants, such as heavy metals, was also demonstrated in the study of Yancheva et al. [114], which tested the properties of aquatic organisms to absorb more of these contaminants in their tissues. An experiment was performed showing that the lysosomal membranes of the cells of these organisms showed a change in stability depending on the dose of cadmium and the duration of administration when large amounts were ingested.

Our research [4] was carried out by examining a large number of carp muscle samples delivered from the water bodies studied at that time in order to determine the amounts of heavy metals in those parts of fish that are used for consumption, not only by many animal species, but also by humans.

It is no coincidence that Ovcharitsa Dam was also present among the studied water bodies, which is characterized by a strong anthropogenic influence due to its proximity to the TPP of Maritsa–East 2. In order to obtain a real picture of the changes in the quantities of the studied heavy metals and the subsequent changes in the state of this water body over time, an analysis of the muscle tissue of carp inhabiting the dam was performed in 2014.

### 5.1. Content of Zn in the Muscle Tissue of Common Carp from the Studied Waters of the Territory of Stara Zagora Region, Bulgaria

Zinc (Zn) is a nonprotein component of strategically important enzyme complexes, which is why deficiency is more common than an excess of it in the bodies of studied aquatic organisms. This determines the lack of restrictions on its content in the muscle tissue of fish in the current regulations—European Regulation 1881 [24] and its subordinate Regulation № 5 of Bulgarian legislation [25]. Like Cu, zinc did not meet the MAC set by Regulation 31 [23].

In the course of our own research, we found that the concentrations of zinc measured in the muscle tissue of carp inhabiting the Chirpan Dam and the Tundzha River in the village of Yagoda were characterized by the highest values for the period of 2010, although they were far from the limits defined in Regulation 31 for the content of the maximum permissible amounts of contaminants in food. The highest concentration was found in the samples from Chirpan Dam and the lowest in Zhrebchevo Dam, which were, respectively, 71 and 89% below the established norms [4].

The trend established in 2010 for the content of Zn in the studied species of fish was maintained in 2011. The established levels of this metal in the samples from all the studied water bodies did not exceed the MAC determined by Regulation 31, 83.8% under the regulation. Pastren Dam was characterized by the lowest measured concentrations (94.6% below the MAC). The results obtained are in full agreement with worldwide research on the zinc content in fish muscle tissue. The symmetry regarding Zn levels in carp muscle tissue from all the mentioned studies is clear evidence that muscle tissue is characterized by low levels of such metals due to their accumulation in organs such as the liver, ovaries, testicles, gills, and kidneys [115,116].

In order to establish the actual state of the ecosystem of the Ovcharitsa Dam as a water body subjected to strong anthropogenic impact, an analysis of carp samples delivered in 2014 was performed. The levels of Zn measured during this period were much lower than the permissible concentrations in Regulation 31 that were enforced at the time (85.3% below this norm) [4].

During the analysis of the results of the three years of research (2010, 2011, and 2014), we found very slight fluctuations in the values of Zn in the muscle tissue of carp from Ovcharitsa Dam. In the period of 2010–2011, an increase of 22.3% was registered, followed by a decrease of 9% in 2014. The established concentrations in these years of research were far below the requirements of the normative documents of the Bulgarian legislation from this period, i.e., there was a symmetry between the registered concentrations for zinc in the three years of research [4].

In their study, Karadede et al. [117] recorded Zn average concentrations of about 8 mg/kg and peak values of 11 mg/kg, which they found in freshwater fish muscle tissue; they were considered low compared to international requirements and did not pose a risk to human consumption.

### 5.2. Content of Pb in the Muscle Tissue of Common Carp from the Studied Water Bodies of the Territory of Stara Zagora Region, Bulgaria

Lead is a heavy metal that has the ability to have a multifaceted toxic effect, even in low doses [118,119]. The reason for this lies in its place in the periodic table, which is determined by its atomic weight and the chemical behavior characteristics of heavy metals. This necessitates the analytical determination of its quantities in the muscle tissue of the studied aquatic organisms. In our study, the accumulating role of the liver was once again observed. In this situation, it was quite normal that, in the muscle tissue of the same organisms that inhabited the studied water bodies in the period of May–December 2010, the measured amounts of lead were much lower than this central organ of metabolism.

However, contrary to expectations, the study conducted in 2010 on the levels of Pb in the muscle tissue of the studied fish and mussels unfortunately found multiple exceedances of the MRL for lead, as regulated in Regulation 31 of 29 July 2004 on the maximum levels of pollutants in food, as well as in the current Ordinance № 5, which corresponds to EU Regulation № 1881 of 2006 with the amendment of 2010 to determine the maximum permissible concentrations of certain contaminants in food. The analysis of the samples showed that the highest levels of lead were registered in the carp of the Koprinka, Zhrebchevo, and Ovcharitsa dams. Priority was given to the levels of Pb at Koprinka Dam, which exceeded the MAC in the above-mentioned normative documents by 1.51 and 1.41 mg/kg, respectively, i.e., by six to nine times. The lowest concentration was reported in the fish from the Tundzha River in the village of Yagoda (twice higher than the regulated MAC). There was an asymmetry between the results we obtained and most of the worldwide studies, which could definitely be a reason for risk in consumption.

The powerful toxic effect that lead has on living organisms [120,121] is the reason why it is present in all international documents relating to its contents in the muscle tissues of fish and mussels, which are often the subject of human consumption. The recorded high values of Pb in all the tested samples clearly showed the presence of pollution in the studied ecosystems during this period. The established symmetry regarding the muscle data and the obtained ones related to the lead content in the livers of carp from the Zagorka, Ovcharitsa, and Koprinka water bodies during the same study period was a sure marker for the state of these ecosystems [4,19].

The registered data from the period of May–August 2011 clearly showed the presence of very low levels of lead in the muscle tissues of all the studied species of aquatic organisms. The role of the liver as the depot of heavy metals was also proved, which explained the level asymmetry in the muscle tissues of the same organisms. The established amounts of Pb in the muscle samples from all the studied water bodies did not exceed the established norms for lead, as regulated in Regulation 31 of 29 July 2004 [23], as well as the currently applicable Regulation № 5 of Bulgarian legislation [25] and EU Regulation № 1881 of 2006 with the amendment from 2010 to determine the maximum permitted concentrations of certain contaminants in foodstuffs [24]. The reported levels were very low, but nevertheless, the highest values were characterized by carp from Pastren Dam (80% below the norms), and the lowest were those from Ovcharitsa Dam (95% below the MAC).

Contrary to our expectations, the concentrations of lead measured in the livers and muscle tissues of the studied species of aquatic organisms inhabiting Ovcharitsa Dam in 2011 were many times lower than those established in 2010. We most likely reported instantaneous contamination with this metal during the past period. Not without significance is the fact that each individual personally accumulates pollutants depending on the abiotic factors of the environment and their own metabolism [4,19].

In order to confirm the thesis about the quantities of lead, it was necessary to perform an analysis of samples from the same water body in 2014, which established the absence of the serious pollution of this hydroecosystem with lead. The results showed that the concentrations in the muscle tissue of the carp from the Ovcharitsa Dam increased compared to 2011 by 81.3%, which did not bring them close to the established norms. The measured levels of lead in this period were below the requirements of the Bulgarian and European legislation by 84%.

The carp samples taken from Ovcharitsa Dam in 2010, 2011, and 2014 showed the probable momentary pollution of the water body in 2010, followed by a significant decrease in levels in 2011 and 2014. The reasons for the values of Pb registered in the muscle tissue in 2010 could be of different nature—from the presence of this toxic metal in water and food intake to the unique metabolism of each individual [4,19].

Comparing the data on the content of Pb in the muscle tissue of carp that inhabited the above water bodies in 2010, 2011, and 2014 a high degree of reliability was found (*p* < 0.001).

The interpretation of the results of all the study periods led to the conclusion that far lower concentrations of heavy metals accumulated in the muscle tissue of fish than in the liver. Most likely, the latter organ, due to its excellent blood supply and high metabolic load, accumulates in its hepatocytes most of the heavy elements. Irrefutable proof of this is the reported values in the muscle samples of the studied aquatic organisms, where exceeding the limit concentration was observed only for lead. In the analysis of meat from *Cyprinus carpio*, Sahiti et al. [122] recorded significantly lower levels of Cd, Pb, Ni, and Cu compared to those in the high metabolically active liver and gills. Studies such as this, as well as a number of others that have analyzed various organs and tissues, including muscle tissue, have shown that this is not a place for the accumulation of heavy metals [101,123,124,125]. From this point of view, the muscle tissue is covered on the outside with skin, which prevents direct contact with the environment. According to El-Moselhy et al. [116], another significant reason for the observed asymmetry regarding the concentrations of these elements in muscle tissue and biochemically active organs is that muscle tissue does not play a role in the detoxification process. The lowest level of accumulation in all the organs and tissues is assigned to blood plasma. In fact, blood serves as a carrier of heavy metals to other tissues and organs.

Given the importance of the muscle tissues of hydrobionts from both biochemical and dietary points of view, this has long been of interest to scientists around the world.

Jaber et al. [126], examining the muscle tissue of carp of the species *Cyprinus carpio*, found low levels of the essential elements of Zn, Cu, and Fe in the muscle tissue of fish, given the fact that these metals are consumed by the body due to their participation as cofactors in many protein systems. In this case, higher concentrations of toxicants, such as lead (1.27 mg/kg), were found in samples from the Tigris River in northern Iraq. In the current regulations concerning the levels of Pb in the muscle tissue of fish, the regulated MAC is 0.3 mg/kg. The presence of levels in the order of 1.27 mg/kg of lead inevitably leads to the formation of oxygen radicals that damage important enzymatic and transport proteins, lipids, and nucleic acids.

Inhabiting the aquatic environment, aquatic organisms are exposed to the complex effects of a significant number of heavy elements [127]. The self-accumulation of individual metals is much lower than accumulation in combination with other metals. This is the reason why, like the liver, a significant number of positive and negative dependencies between individual metals were found in the muscle tissues of the studied hydrobionts.

## 6. Conclusions

The established symmetric and asymmetric dependencies in the contents of the heavy metals of lead and zinc in the studied water bodies and aquabionts showed the following.

During our research, symmetry was found regarding the levels of Zn in the sediments of the Ovcharitsa and Zhrebchevo dams, which were twice the MPC for arable land (Regulation № 3 of Bulgarian legislation). Symmetry was also observed between the results for Zn and Pb in the studied sediments, as again the “favorites” in terms of content were the samples delivered from Zhrebchevo Dam and, especially, from Ovcharitsa Dam. In the case of lead, however, the norms for soils established in Bulgaria were not exceeded.

During the analysis of the data on the quantities of metals in the sediments of the water bodies we studied, symmetry was found in the contents of these elements at the four points along the Tundzha River, which suggested at least two facts:Heavy metal pollution was in the section between the village of Yagoda and the town of Nikolaevo, probably from recent mining in the Tvarditsa Region;Zhrebchevo Dam served as a precipitator of heavy metals (and probably other pollutants), thus purifying the Tundzha River.

As a result of our research to determine the amounts of metals in algae and higher aquatic plants, symmetry was found when comparing the high values, among which the samples of water buttercup (Pb at 10.57 mg/kg) and algae samples from the Tundzha River after the wall of the Zhrebchevo Dam (8.02 mg/kg) and the Sazliyka River (Pb at 9.26 mg/kg) stood out. The analysis of the muscles and livers of the studied fish showed an asymmetry in the accumulation of zinc, and this process was more intense in the liver. Symmetry was observed in this biochemical process for Pb in the liver and the muscle tissue of the carp from the water bodies we studied.

By means of a comparison between the results of our own research and others conducted worldwide, the thesis was confirmed that the sediments of water bodies, by “sealing” the pasts of water bodies, could serve as good ecological markers of heavy metal pollution.

The comparison made between the studies carried out in the Stara Zagora region, Bulgaria, and other countries confirmed the abilities of the plant and animal representatives of the hydrobionts to accumulate heavy metals and to be excellent bioindicators of pollution with these toxicants.

Considering the processes of global environmental pollution, it is necessary to continue research in this area with other environmental markers and analysis parameters. Assessments of ecological status by determining the contents of other toxic metals (such as cadmium, mercury, chromium, nickel, etc.) in hydroecosystems and the hydrobionts inhabiting them are recommended avenues for future research.

## Figures and Tables

**Figure 1 ijerph-19-09600-f001:**
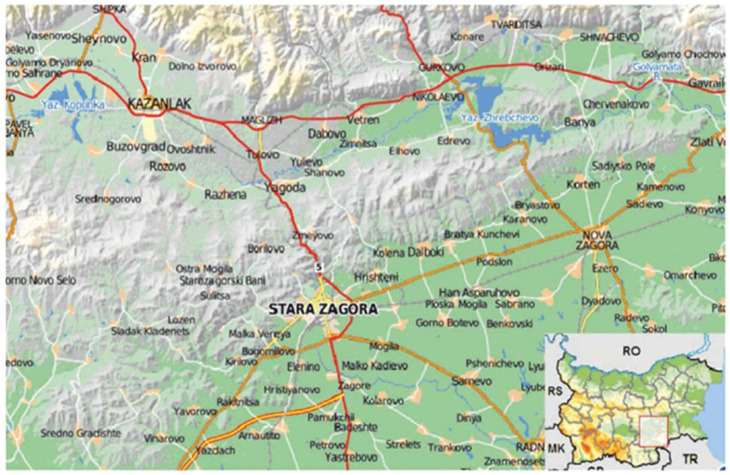
Map of Stara Zagora region, Bulgaria.

**Figure 2 ijerph-19-09600-f002:**
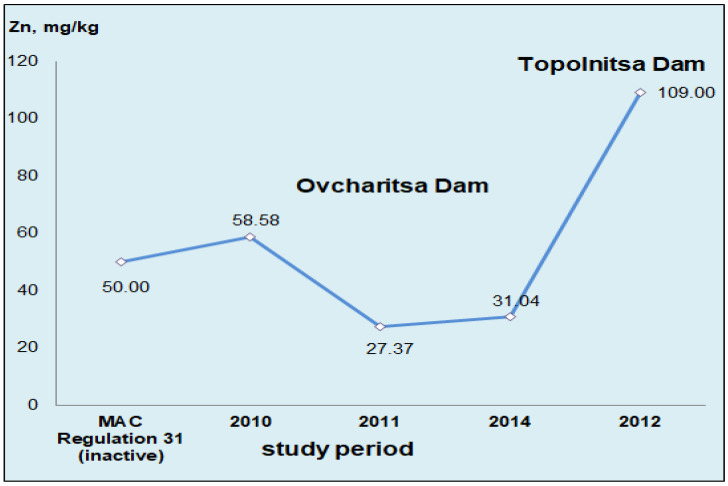
Content of Zn in liver samples of common carp inhabiting Ovcharitsa Dam in the period of 2010–2014 [4], as well as Topolnitsa Dam in 2012 [101].

**Figure 3 ijerph-19-09600-f003:**
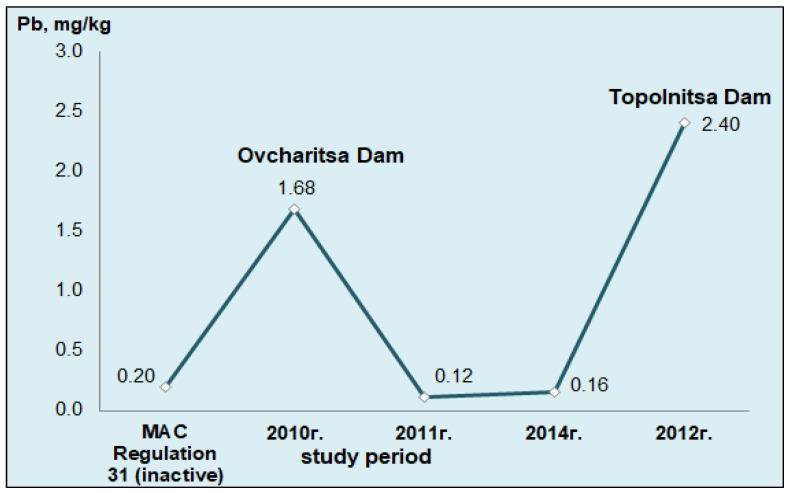
Content of Pb in liver samples of common carp inhabiting Ovcharitsa Dam in the period of 2010–2014 [4], as well as Topolnitsa Dam in 2012 [101].

**Table 1 ijerph-19-09600-t001:** Norms for maximum allowable concentrations of heavy metals Pb and Zn in arable land under Regulation № 3 of the 1 August 2008 Bulgarian legislation.

Heavy Metal	pH (H_2_O)	Arable Land
lead (Pb)	6.0–7.4	100
zinc (Zn)	6.0–7.4	320

**Table 2 ijerph-19-09600-t002:** Quality standards for the heavy metals lead and zinc in freshwater fish (under Regulation 31 of 29 July 2004 on the maximum permissible levels of contaminants in food).

Chemical	Food	Norm (mg/kg of Fresh Product)
Element		
lead	Fish meat (when the fish is intended to be consumed whole, this refers to the whole fish)	0.2
zinc	Freshwater fish	50

**Table 3 ijerph-19-09600-t003:** Quality standards for Pb in fish meat under EU Regulation № 1881 of 2006, as amended in 2010, and the corresponding Regulation № 5 in Bulgarian legislation, which entered into force on 9 February 2015, to determine the maximum permitted concentrations of some pollutants in foodstuffs.

Chemical	Food	The Norm
Element		(mg/kg Wet Weight)
lead	Fish meat	0.3

## Data Availability

Not applicable.

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
