# Peer review of "Content of Pb and Zn in Sediments and Hydrobionts as Ecological Markers for Pollution Assessment of Freshwater Objects in Bulgaria—A Review"

_ijerph, 2022, doi:10.3390/ijerph19159600_

Round 1
Reviewer 1 Report
The reviewed manuscript, "Dynamics in Some Ecological-Biochemical Parameters of the Freshwater Objects Assessment for Pb and Zn Pollution in Bulgaria", is exciting and informative. This study aims to describe the trends in specific ecological-biochemical characteristics of freshwater assessment objects for Pb and Zn contamination in Bulgaria while comparing the data to other nations and areas.
Heavy metals such as Cu, Zn, Pb, Cd, Ni, Co, and others are found in a significant number of waste products and household products, industry, and agriculture, indicating a high level of risk of contamination of aquatic ecosystems with these elements and subsequent metabolic changes in the bodies of aquatic organisms while also causing significant economic losses. This is a good topic to cover in a review article.
Also, the authors have compared and reviewed several reported pieces of literature to support their observations. Therefore, the cited references are appropriate and current. The review's conclusion is reasonable, and the authors have reached the final verdict using in-depth discussion. However, I have specific comments that need to be addressed before the manuscript can be accepted for publication.
a. If there is a word limit issue, sub-section 1.1 and 1.2 can be reduced. Also suggested shortening the hydrations and humans as consumers section from the introduction.
b. The map is not clear. However, the high-res photo will be appreciable.
c. The mechanism of ecological markers should be discussed in detail, preferably with schematics.
d. Remove the word "Some" from the title and document.
e. In figure 3, the timeline is 2010-2014. Can it be extended up to 2022?
Author Response
- A) Sections 1.2 and 1.5 have been reduced.
- B) The picture in fig. 1 has been replaced by another one that is clearer and has a higher resolution.
- C) The mechanism of ecological markers is discussed in detail by the text.
- D) The title has been completely changed with the word "some" removed from it.
- E) In fig. 3 period is 2010-2014 and it cannot be extended, since this particular study ended in 2014. However, data from similar studies for later years are mentioned in the text.
Reviewer 2 Report
Review on Manuscript Number: ijerph-1802668
Dynamics in Some Ecological-Biochemical Parameters of the Freshwater Objects Assessment for Pb and Zn Pollution in Bulgaria–A Review
1. Summary
The manuscript aims to present a summary of the current state of knowledge for Pb and Zn pollution in freshwater water bodies in the territory of Stara Zagora Region, in Bulgaria, based on the evaluation of ecological-biochemical parameters of freshwater assessment objects - sediment, algae and higher aquatic plants, and liver and muscle tissue of common carp. The data from this region are compared to other regions. This region is considered a risk area, because it is heavily impacted by mining activity (Mini Maritsa East Mines EAD operates the biggest lignite coal mine in Bulgaria).
In the first part of the manuscript “Introduction”, the authors provide the framework of the area, and general considerations of (1) pollution of hydro-ecosystems and type of pollutants, (2) legislation in Bulgaria and Europe regulating the permissible limits for heavy metals in hydroecosystems, (3) biological indicators for the condition of the aquatic environment (hydrobionts), (4) Biological species suitable as ecological markers (mussels; fish species - common carp and crucian carp), (5) accumulation of heavy metals and their effect on the organism of hydrobionts and humans as consumers. In the second part of the manuscript, the contents of Zn and Pb in the sediment, algae and higher aquatic plants, liver and musculature of common carp of freshwater water bodies, collected from published studies are discussed. The authors also include a comparison between published studies and their own studies.
2. Overall opinion
The manuscript is of potential interest for researchers and decision makers, but needs a deep revision before publication to clarify and organise the ideas, and provide a better structure and flow. As we go forward in the reading of the manuscript we get confused: a great part of ideas are repeated, the discussion is confusing and it is difficult to follow the evaluation of the data available from existing studies on this topic (part of it is from the authors). At the end, there are no recommendations for future research. The topic under review must be clearly established, and the context of the review must be provided.
The relation that the authors pretend to establish between the results from their own studies (“…our studies…” as frequently is mentioned) with the ones from the mentioned project and published literature need to be clarified when the aim of the study was established (topic of the revision) in the introduction. The authors frequently refer to their own studies (references 4, 18, 19, 20, 21, 22, 27, 31, 32, 36, 85…) and a comparative analysis is present along the second part of the manuscript (Lines 453…
Once the text has been revised and organised, the discussion should be complemented by graphs, it would turn the reading more interesting.
First paragraphs of sections 2 to 5, resemble like a short introduction for each section, but part of the information/discussion was already mentioned in Section 1. The text needs to be reorganised.
The “dynamics” included in the title is not well explained/ established along the discussion.
English format needs revision, regarding verbal forms, sentences, punctuation and correction of typos in the text. Some sentences are confuse, unfinished or include “lost” words.
2. Comments
The Introduction needs to be improved. It should include an overview of the topic, provide some context, and explain why a review of this topic is relevant.
The aim of this review is not properly established: the Abstract mentions “The purpose of this review is to present the dynamics in some ecological-biochemical parameters of freshwater assessment objects for Pb and Zn pollution in Bulgaria, as the data are compared with other countries and regions.”; in Introduction (lines 48-49) it is mentioned “… This is the reason for conducting research on the levels of heavy metals in the hydrobionts inhabiting key water bodies in the Stara Zagora Region.” The topic of this review must be set clearly.
Which ecological-biochemical parameters of freshwater assessment objects were considered? Information on this only appears in line 150.
At the end of the section 1.2 is mentioned the method (AA) considered adequate to determine metals in samples of different nature. This statement needs to be justified, there are other methods also accurate and adequate for specific types of samples. Anyway, the method does not need to be mentioned, unless it is mandatory for the selection of data been discussed. In the following paragraphs, three methods used for monitoring aquatic ecosystems are described (Lines 174-208). This last part, method and methods, is not in line with the previous text of this section, and as little or none support from literature references.
Section 1.4 - Last paragraph repeats information previously mentioned.
Section 1.5 - Zn and Pb effects on the organism of hydrobionts are presented in detail, and several studies are cited. However, last 3 paragraphs (metals in surface waters) are not in line with the content of this section (effects of metals on hydrobionts and humans).
Titles of sections 2 to 5 mention “Content of some heavy metals…” but the study is about Zn and Pb. The sub-titles are repetitive “…from the study water bodies in the territory of Stara Zagora Region, Bulgaria”.
Line 392-393 - “Precipitation depends on the water flow rate and the particle size. This process leads to a decrease in the concentrations of various pollutants (including heavy metals) in the aquatic environment.” – precipitation is not the only process controlling the accumulation, or retention, of metals in sediments, sorption and exchange are examples of other major processes occurring between the metals in solution and sediment.
Line 407 to 408 - “This study proves the generally accepted thesis that in the sediment of the water bodies are concentrated a large part of the metals…” Instead of “proves” consider “is in accordance with”
In Sections 2 to 5, information on other metals than Zn and Pb, are often mentioned. As the review concerns to Zn and Pb, the focus should be kept onto these two metals. Examples: Line 422 to 424 (Cd); 582; 605; 616;
Line 427-426 – this has already been mentioned
Line 436 - “The results of the studies on a global scale and on the territory of Bulgaria determined…” – which studies on a global scale?
Paragraphs in lines 436 to 442 and 445 to 450 refer to the same subject, which is fragmented by sentence in Line 443-444. Is this out of place?
Line 493 to 501– this has already been mentioned in the Introduction…
Line 502 - Two studies confirm?
Sentences like “Lead (Pb), as a typical heavy metal, …” can not appear in a review manuscript.
Line 572 – “In the world literature…” – literature
Figure 2, 3 – As they are shown, do not illustrate the text “The quantities of this heavy metal were compared with the results obtained in our study in the period 2010 - 2014, which showed that the accumulation in common carp inhabiting the Topolnitsa Dam in 2012 (Fig. 2) is significantly higher and exceeds twice the requirements of the then active in Bulgarian Legislation Regulation 31 [23].” And “Compared to our data from the research of carp liver samples, inhabiting Ovcharitsa Dam in the period 2010 - 2014 (2010 - 1.68 mg/kg), these results are several times higher (Fig.3).”
The conclusions are related with authors own research and not with the literature review proposed at the beginning of the manuscript:
“The established symmetric and asymmetric dependences in the contents of the heavy metals lead and zinc in the studied water bodies and aquabions show the following: During our research…; During the analysis of the data on the quantities of metals in the sediment of the water bodies we studied, symmetry…; As a result of our research to determine the amounts of metals in algae and higher aquatic plants, symmetry was found…; The analysis of the muscles and liver of the studied fish shows an asymmetry in the accumulation of zinc…”
Abbreviations should be defined the first time they appear in each of three sections: the abstract; the main text; the first figure or table. When defined for the first time, the abbreviation should be added in parentheses after the written-out form.
Figure 1 needs to be improved
Line 123 “…of the heavy metals (Zn и Pb).”
Zinc, not zink (table 2)
Tables are not mentioned in the text
Why table 3 only shows Pb?
Consider use muscle tissue instead of musculature
Author Response
- The topic of the manuscript is clearly stated.
- The title of the manuscript and the stated objective have been completely changed. In sections 2 to 5, both own research on the topic and those on a global scale are indicated and commented on, with the data from the own studies being presented in more detail.
- The manuscript contains 2 graphs to compare the data obtained by us in the study period 2010-2014 with those of Yancheva et al. (2014) since 2012. Due to the fact that the manuscript is a review, the results of other studies are indicated and discussed, but without being illustrated with graphs.
- The text in the first paragraphs of sections 2 to 5 has been reduced and reorganized.
- The title of the manuscript has been changed and explained in the context of the review.
- Some of the sentences in the text have been edited.
- The purpose of the review has been changed and correctly established, with a reduction of part of the introduction.
- The ecological markers for the assessment of the pollution of the studied water bodies are clearly indicated and explained in all sections of the manuscript.
- On old line 149, the object of research is indicated - Cyprinus carpio, L., and this object is indicated in all sections of the manuscript.
- Tables 2 and 3 have been corrected, stating the established limits only for the investigated metals – Pb and Zn in fish, and the titles of the tables have been changed.
- Several methods exist that are accurate and adequate in determining the amounts of metals in specific types of samples. We have indicated that one of them is atomic absorption, because we had the appropriate equipment for this method at the time of the study, and it is suitable for this type of sample.
- Strategic methods for monitoring aquatic ecosystems were removed due to lack of consistency with the previous text of this section.
- Section 1.4 has been reduced. The object of study - common carp (Cyprinus carpio, L.) - is described, and the advantages of using this type of fish for the study are indicated.
- The last two paragraphs of section 1.5 have been removed due to lack of connection with the content of this section.
- Changed the headings of sections 2 to 5, indicating only the studied metals lead and zinc.
- The text related to the influence of various processes on the accumulation of heavy metals in the sediment has been reduced. The recommendation to add the text of the sorption and ion exchange processes has been implemented.
- Corrected the text of old lines 407 and 408.
- Corrected radles 2 to 5, keeping the focus on the metals Pb and Zn.
- Removed repeated the text from old lines 426 and 427.
- It has been specified which studies are in question in line 436.
- The text of old lines 436 to 442 and 445 to 450 has been clarified and revised.
- The text of old lines 493 to 501 has been shortened.
- Both studies mentioned on old line 502 confirm the statement made.
- The opening sentence of subsection "Pb content in algae and aquatic plants inhabiting freshwater water bodies in the Stara Zagora Region, Bulgaria" has been corrected.
- On old line 572, the expression "In world literature..." was replaced with the expression "In literature".
- Abbreviations are defined the first time they appear in the text of all sections.
- A correction was made to old line 123.
- Corrected the name of the zinc element in table 2.
- In table 3, only the limits for lead are indicated, since only such are present in the current regulatory documents. Permissible norms are not specified for zinc.
- Throughout the text, the word "musculature" has been replaced by "muscle tissue".
- Text was added to the conclusions referring to research by other authors.
- At the end of the manuscript, recommendations for future research are added.
Reviewer 3 Report
Comments to the Article “Dynamics in some ecological-biochemical parameters of the freshwater objects assessment for Pb and Zn pollution in Bulgaria – a review”, by Elica Bogomilova Valkova et al.
The authors in this review present the dynamics in ecological/biochemical parameters for heavy metal water pollution.
The article contains the sections Introduction and Conclusion and between presents the results of the developed analysis.
In my opinion, the text needs a careful revision by the authors.
· The introduction section is quite extensive and confusing. It does not only present the main goals of the study, but also mentions European and Bulgarian legislation, methods for monitoring aquatics ecosystems (although this text is included in the subsection of legislation), and the effects of heavy metal accumulation on the organism of humans and hydrobionts.
· Tables 1 to 3 are in the Introduction section with no reference in the text to them.
· No detailed description of the methods used to measure heavy metal concentrations are presented.
· The sentence starting in line 180 is not understandable.
· At the beginning of line 249, is it Cammon carp or common?
· The sentences in lines 397 - 400 seems to be contradictory. And the same sentence is repeated in line 427.
· Figures 2 and 3 are misleading, in the sense that they include information of two different sources in the same graphic. Authors should consider an alternative way to include the information about Topolnitsa Dam.
· Several times in the text, the authors mention that Bulgarian Legislation Regulation 31 is no longer active. Consider stating this only once and refer that it is valid for all the study.
Author Response
- The introductory section has been shortened by also removing the text on the methods of monitoring aquatic ecosystems.
- References are made in the text to Tables 1 to 3.
- The manuscript does not provide a detailed description of the methods for measuring the amounts of the heavy metals studied, but only mentions the atomic absorption method, due to the fact that according to editor 1, such a description of the method is not necessary, unless it is important for the selection of the studies of other authors.
- The sentence from old line 180 is removed.
- A correction was made to the titles of figures 2 and 3, where it is clearly visible which data are compared with which.
- Only once in the manuscript of line 146 is it mentioned that the commented Regulation 31 of the Bulgarian legislation is no longer effective.
Round 2
Reviewer 2 Report
Comments on the Revised Version (ijerph-1802668-peer-review-v3)
Dynamics in Some Ecological-Biochemical Parameters of the Freshwater Objects Assessment for Pb and Zn Pollution in Bulgaria–A Review
In this revised version important amendments were made. Nevertheless, in my opinion the manuscript can not be published in the present format.
I reiterate that this study is interesting, and justifies to be published, but it is not yet ready.
I recommend the authors to read the manuscript with careful to organize better the text. The English format needs to be revised (I recommend a revision of the English format by a native speaker). A few examples are:
- Lines 57-58 “In a significant number of waste products and household products, industry and agriculture are contained heavy metals such as Cu, Zn, Pb, Cd, Ni, Co, etc.,…” – are contained (?)… such as… etc…(?)
- Lead (Pb) and Zinc (Zn). At the beginning of the sentences is Lead or Zinc, but in the sentences is Pb or Zn. (example: Lines 123, 325, 639, 644, 702…)
- The sub-titles of sections 2 to 5 are repetitive “…from the study water bodies in the territory of Stara Zagora Region, Bulgaria”
Major recommendations:
- Aim “The purpose of this review is to present the content of Pb and Zn in sediments and hydrobionts as ecological markers…” – consider to use describe/analysis instead of “presents the content”.
- In the Section 1 (Introduction), before the start of section 1.1, the topic under review must be clearly established, and why is relevant to present a review on this topic. Clearly state which ecological-biochemical parameters of freshwater assessment objects were considered. Also, an overview of the review should be given, and the relation between the results from the authors own studies with the ones from the project that is mentioned (Line 385) described in the second part of the manuscript should be put into the context of the topic.
- At the end, a poor recommendation for future research was added (“Considering the processes of global warming and environmental pollution, it is necessary to continue the research in this area with other dangerous heavy metals (such as cadmium, chromium, nickel, etc.”). A review article must indicate the best avenues for future research (recommendations about ecological markers, parameters to be analysed, etc).
From the first review:
Line 123 “…of the heavy metals (Zn и Pb).”
Line 454 - “Lead (Pb), as a typical heavy metal, …” this type of sentence can not appear in a review manuscript.
Author Response
Comments on the Revised Version (ijerph-1802668-peer-review-v3)
Dynamics in Some Ecological-Biochemical Parameters of the Freshwater Objects Assessment for Pb and Zn Pollution in Bulgaria–A Review
Response: The title of the review has been changed. The title is: “Content of Pb and Zn in sediments and hydrobionts as ecological markers for pollution assessment of freshwater objects in Bulgaria - a Review”.
In this revised version important amendments were made. Nevertheless, in my opinion the manuscript cannot be published in the present format.
I reiterate that this study is interesting, and justifies to be published, but it is not yet ready.
I recommend the authors to read the manuscript with careful to organize better the text. The English format needs to be revised (I recommend a revision of the English format by a native speaker). A few examples are:
- Lines 57-58 “In a significant number of waste products and household products, industry and agriculture are contained heavy metals such as Cu, Zn, Pb, Cd, Ni, Co, etc.,” – are contained (?)… such as… etc…(?)
Response: The text was reorganized by a native English speaker.
- Lead (Pb) and Zinc (Zn). At the beginning of the sentences is Lead or Zinc, but in the sentences is Pb or Zn. (example: Lines 123, 325, 639, 644, 702…)
Response: Some of the sentences in the text have been edited.
- The sub-titles of sections 2 to 5 are repetitive “…from the study water bodies in the territory of Stara Zagora Region, Bulgaria”
Response: In the subheadings, it is indicated "from the studied water bodies in the territory of the Stara Zagora region, Bulgaria", as it concerns the results of our research precisely on these water bodies. The information from line 330 to line 375 concerns other research on the subject.
Major recommendations:
- Aim “The purpose of this review is to present the content of Pb and Zn in sediments and hydrobionts as ecological markers…” – consider to use describe/analysis instead of “presents the content”.
Response: The aim of review was changed.
- In the Section 1 (Introduction), before the start of section 1.1, the topic under review must be clearly established, and why is relevant to present a review on this topic. Clearly state which ecological-biochemical parameters of freshwater assessment objects were considered. Also, an overview of the review should be given, and the relation between the results from the authors own studies with the ones from the project that is mentioned (Line 385) described in the second part of the manuscript should be put into the context of the topic.
Response: The text in these zone was revised.
- At the end, a poor recommendation for future research was added (“Considering the processes of global warming and environmental pollution, it is necessary to continue the research in this area with other dangerous heavy metals (such as cadmium, chromium, nickel, etc.”). A review article must indicate the best avenues for future research (recommendations about ecological markers, parameters to be analysed, etc).
Response: The recommendation for future research was revised.
From the first review:
Line 123 “…of the heavy metals (Zn и Pb).”
Response: Some of the sentences in the text have been edited.
Line 454 - “Lead (Pb), as a typical heavy metal, …” this type of sentence can not appear in a review manuscript.
Response: The text was revised.
*The title of the review has been changed. The title is: “Content of Pb and Zn in sediments and hydrobionts as ecological markers for pollution assessment of freshwater objects in Bulgaria - a Review”.
*Added new authors in the reference.